# The Divergence of Chromosome Structures and 45S Ribosomal DNA Organization in *Cucumis debilis* Inferred by Comparative Molecular Cytogenetic Mapping

**DOI:** 10.3390/plants11151960

**Published:** 2022-07-28

**Authors:** Agus Budi Setiawan, Aziz Purwantoro, Chee How Teo, Phan Thi Phuong Nhi, Kenji Kato, Shinji Kikuchi, Takato Koba

**Affiliations:** 1Laboratory of Plant Breeding, Department of Agronomy, Faculty of Agriculture, Universitas Gadjah Mada, Jalan Flora, Bulaksumur, Yogyakarta 55281, Indonesia; azizp@ugm.ac.id; 2Centre for Research in Biotechnology for Agriculture, University of Malaya, Kuala Lumpur 50603, Malaysia; cheehow.teo@um.edu.my; 3Faculty of Agronomy, University of Agriculture and Forestry, Hue University, No 102 Phung Hung Street, Hue 530000, Vietnam; phanthiphuongnhi@huaf.edu.vn; 4Graduate School of Environmental and Life Science, Okayama University, Kita-ku 700-8530, Japan; kenkato@okayama-u.ac.jp; 5Laboratory of Genetic and Plant Breeding, Graduate School of Horticulture, Chiba University, Chiba 271-8510, Japan; skikuchi@faculty.chiba-u.jp (S.K.); koba@faculty.chiba-u.jp (T.K.)

**Keywords:** *C. debilis* fruit, *C. debilis* rDNA loci, *Cucumis* chromosomes, divergence of rDNA loci

## Abstract

*Cucumis debilis* W.J.de Wilde & Duyfjes is an annual and monoecious plant. This species is endemic to Southeast Asia, particularly Vietnam. However, *C. debilis* is rarely studied, and no detailed information is available regarding its basic chromosome number, 45S ribosomal DNA (rDNA) status, and divergence among other *Cucumis* species. In this study, we characterized the morphological characters and determined and investigated the basic chromosome number and chromosomal distribution of 45S rDNA of *C. debilis* using the fluorescent in situ hybridization (FISH) technique. A maximum likelihood tree was constructed by combining the chloroplast and internal transcribed spacer of 45S rDNAs to infer its relationship within *Cucumis*. *C. debilis* had an oval fruit shape, green fruit peel, and protrusion-like white spots during the immature fruit stage. FISH analysis using 45S rDNA probe showed three pairs of 45S rDNA loci located at the terminal region in *C. debilis*, similar to *C. hystrix*. Meanwhile, two, two, and five pairs of 45S rDNA loci were observed for *C. melo*, *C. metuliferus*, and *C. sativus*, respectively. One melon (P90) and cucumber accessions exhibited different chromosomal localizations compared with other members of *Cucumis*. The majority of *Cucumis* species showed the terminal location of 45S rDNA, but melon P90 and cucumber exhibited terminal–interstitial and all interstitial orientations of 45S rDNA loci. Based on molecular cytogenetics and phylogenetic evidence, *C. debilis* is more closely related to cucumber than melon. Therefore, *C. debilis* may serve as a potential parental accession for genetic improvement of cucumber through interspecific hybridization.

## 1. Introduction

*Cucumis* is one of the most important genera in Cucurbitaceae, consisting of numerous economically cultivated crops around the world, i.e., *Cucumis sativus* and *C. melo* [1,2,3,4]. Most *Cucumis* species are annuals, whereas *C. picrocarpus*, *C. trigonus*, *C. heptadactylus*, *C. africanus*, *C. myriocarpus*, *C. prophetarum*, *C. zeyheri*, *C. messorius*, and *C. sagittatus* are perennial plants [4]. Kirkbride [5] proposed a comprehensive taxonomical classification of *Cucumis* species based on their phenotypic characters, basic chromosome numbers, and geographic origins. However, his taxonomical classification was challenged with the discovery of *C. hystrix*, which is cross-compatible with *C. sativus* [6,7]. The interspecific hybridization between them resulted in an amphidiploid species (2n = 4x = 38), which may be further used for crop improvement and to broaden the genetic base of cucumber through backcross introgression lines [8,9,10].

The genus *Cucumis* is represented by 32 species [5], but the number is increasing with the discovery of new species and/or re-identification of existing species belonging to other genera [4,11,12,13,14,15,16]. Most *Cucumis* species originated from Africa, whereas several were from Australia/New Guinea (*C. variabilis*, *Cucumis* sp. nov. A, *C. villosior*, *C. umbellatus*, *C. queenslandicus*, and *C. picrocarpus*) and Asia (*C. melo*, *C. sativus*, *C. hystrix*, and *C. debilis*) [14]. *C. debilis* W.J.de Wilde & Duyfjes, a species from Southeast Asia, is endemic to Vietnam and was first discovered by De Wilde and Duyfjes [13] from one of their existing collections in 1931 in Paris Herbarium. *C. debilis* is an annual plant and closely related to *C. sativus* compared with *C. melo* based on the phylogenetic analysis of chloroplast and nuclear internal transcribed spacer (ITS) sequences [4,14,16,17,18].

*Cucumis* species harbor highly conserved and tandemly repeated sequences of 45S ribosomal DNA (rDNA) at the nucleolar organizer regions [19,20,21,22]. The 18S, 5.8S, and 28S ribosomal RNAs (rRNAs), which are produced via a common transcription unit of 45S rDNA, are responsible for ribosome biogenesis and confer various functions in plant development and responses to biotic and abiotic stresses [23,24]. The copy number and intergenic spacer sequence of the 45S rDNA and the number of 45S rDNA loci in chromosomes are highly variable in plants [21,25,26,27,28,29,30,31,32,33,34,35,36,37,38,39,40,41,42,43,44,45,46]. Although the chromosomal organization of 45S rDNA has been extensively studied in *Cucumis* species, and the related information has been deposited in the plant rDNA database [19,20,22,47,48,49,50,51,52,53], no information is available in relation to the basic chromosome number and chromosomal distribution of 45S rDNA in *C. debilis*. This study was conducted to characterize the morphology and to investigate the molecular cytogenetic information of *C. debilis*, which has a particularly different basic chromosome number and chromosomal organization of 45S rDNA gene compared to other *Cucumis* species. Here, we confirmed the basic chromosome number of *C. debilis* and chromosomal localization of 45S rDNA among *Cucumis* species using a fluorescent in situ hybridization (FISH) technique. In addition, the relationship of *C. debilis* within *Cucumis* was inferred from a maximum likelihood tree of chloroplast and ITS sequences.

## 2. Results

### 2.1. Plant Morphology of C. debilis

*C. debilis* is a monoecious and annual plant originating from Vietnam. This species has 24 chromosomes and six loci of 45S rDNA (Figure 1a–c). This species is an herbaceous plant with a slender stem, sparsely aculeate tendrils, hairy stem, deeply cordate leaf base, broad triangular leaf lobes with spiny hairs, and acuminate leaf apex (Figure 1d–f). The fruit shape is oval, and the fruit peel of the immature stage has green color and is marked with protrusion-like white spots. The fruit starts the breaker stage at the early mature phase, which involves a definite break in color from green to slightly yellow, indicating the initiation of the ripening process (Figure 1e). The *C. debilis* plant cultivated in the greenhouse of Okayama University failed to develop flowers and fruits (Figure 1f).

### 2.2. Basic Chromosome Number and Number of 45S rDNA Loci in Cucumis Species

Chromosome counting of six *Cucumis* species showed that *C. debilis* (Figure 1a), *C. hystrix* (Figure 2a), *C. metuliferus* (Figure 2g), and *C. melo* (Figure 2j) have 2n = 2x = 24 chromosomes, whereas *C. sativus* has 2n = 2x = 14 chromosomes (Figure 2d). Therefore, cucumber is the only species with 14 chromosomes among the *Cucumis* species investigated in this study. Physical mapping of 45S rDNA was conducted on the mitotic and meiotic chromosomes of *Cucumis* using the FISH technique. FISH results showed that *C. debilis* has three pairs of 45S rDNA loci with two major and one minor loci (Table 1 and Figure 1b,c), whereas *C. hystix* and *C. sativus* have six (four major and two minor) and ten (six major and four minor) loci of 45S rDNA, respectively (Table 2, Figure 2b,c,e,f). Meanwhile, *C. metuliferus* has two major loci of 45S rDNA (Table 1 and Figure 2h,i). Two accessions of melon, P90 and US205, have two pairs of 45S rDNA loci with different signal intensities (Table 1, Figure 2k,l,n,o). The metaphase chromosomes of P90 consisted of one pair of major and minor loci of each 45S rDNA, whereas pachytene of US205 consisted of two major loci of 45S rDNA. These results suggest that 45S rDNA are highly conserved in *Cucumis* species, including *C. debilis*, with a high variation in the number of 45S rDNA gene loci.

### 2.3. Chromosomal Organization of 45S rDNA in Cucumis Species

Most of the *Cucumis* species used in this study showed the terminal type of 45S rDNA organization except for *C. sativus* and *C. melo* (P90) (Figure 2). *C. debilis* and *C. hystrix* have six loci of terminal type 45S rDNA (Figure 1c or Figure 2c, respectively). The 45S rDNA probe was hybridized to the terminal regions of two pairs of metaphase chromosomes in *C. metuliferus* and a pair of pachytene chromosomes in *C. melo* (US205) (Figure 2i,o). All the 45S rDNA loci in *C. sativus* were located in the interstitial region of the chromosomes (Figure 2f). Two melon accessions showed different chromosomal organization of the 45S rDNA. The 45S rDNA was located at the terminal and interstitial regions in P90 (Figure 2l), whereas in US205, the signals were only detected at the terminal regions (Figure 2o). These results suggest that the high variation of 45S rDNA in *Cucumis* occurred not only in their total locus number, but also in their chromosomal locations.

### 2.4. Clustering Analysis and Ideogram of Cucumis Species

A maximum likelihood tree showed that *Cucumis* species is divided into two clusters, i.e., *C. sativus* and *C. melo* groups (Figure 3). The phylogenetic tree revealed that *C. debilis* is closely related to *C. sativus* and *C. hystrix* compared with *C. melo*, confirming the findings of previous studies [4,14,16,17,18]. Meanwhile, *C. melo* and *C. metuliferus* were grouped in one cluster. These cluster analysis results are consistent with the ideogram (Figure 3), in which species having more than four loci of 45S rDNA are grouped into one cluster, namely *C. hystrix, C. sativus,* and *C. debilis*. Meanwhile, species having just four 45S rDNA loci, such as *C. metuliferus* and *C. melo*, were separated into distinct groups. The location variation of 45S rDNA in *Cucumis* chromosomes investigated in this work indicates a preference for terminal rather than the interstitial location of 45S rDNA. *C. hystrix, C. debilis, C. metuliferus,* and one accession of *C. melo* (US205) had terminal 45S rDNA loci, but *C. sativus* had all 45S rDNA loci in the interstitial region, and *C. melo* (P90) had both terminal and interstitial 45S rDNA loci.

## 3. Discussion

Currently, 65 species of *Cucumis* exist throughout the world, and the center of origin for *Cucumis* species can be divided into three regions based on their geographic locations, i.e., Africa, Australia/New Guinea, and Asia [4,14]. *C. debilis* is native to Vietnam [13]. *C. debilis* was first determined by De Wilde and Duyfjes [13], and its plant morphology, particularly the leaves, tendrils, and flowers, was finely described. However, the fruit morphology has not been described well, and the information on its molecular cytogenetics, particularly its basic chromosome number and 45S rDNA status, was limited. De Wilde et al. [54] attempted to grow the plant in the Netherlands, but it failed to develop flowers and fruits. Their trial results coincided with our trials in Japan, where the initiation of the plant into the generative stage and production of normal flowers and fruits were difficult. Several wild types of *Cucumis* species or landraces exhibit specific adaptations to environmental conditions of their origin region [55]. This result suggests that *C. debilis* requires a specific agroclimate to enter the generative phase and develop normal fruits when it is introduced into a new location. Further studies are required to overcome this issue by investigating the effect of modified environmental conditions that may affect the growth and development of *C. debilis* fruit.

In this study, we showed that *C. debilis* has 24 chromosomes (Figure 1a), similar to the majority of *Cucumis* species [19,49]. All diploid *Cucumis* species used in this study have 24 chromosomes, except cucumber, which has 14 chromosomes [21,22]. Three pairs of 45S rDNA loci were detected at the terminal regions of *C. debilis* chromosomes (Figure 1c). This result was coincident with the chromosomal distribution of 45S rDNA in *Cucumis* chromosomes, which are mainly located in the terminal region [22]. Most of the *Cucumis* species used in this study also showed the terminal type of 45S rDNA chromosomal organization (Figure 1 and Figure 2). The terminal locations of 45S rDNA in these species might have been caused by homologous recombination events, where the rDNAs may be subject to allelic and non-allelic recombination [56]. However, two different chromosomal organizations of 45S rDNA were observed in melon and cucumber. Melon has two types of 45S rDNA locations, i.e., (1) both pairs are located at the terminals and (2) terminal–interstitial directions (Figure 2l and Figure 2o, respectively). Meanwhile, cucumber has all the interstitial locations (Figure 2f). This result was similar to previous findings that the majority of 45S rDNA loci in cucumbers were interstitial, but melon contained two unique types of 45S rDNA loci, all terminals, and terminal–interstitial [19,20,21,22,47]. In addition, cucumber has a wide variation in the number of 45S rDNA, ranging from 8 to 10 loci depending on the cultivar [20,47].

The interstitial type of 45S rDNA organization is only found in melon P90 (Figure 2l) and cucumber (Figure 2f). One pair of chromosomes in melon accession P90 (Figure 2l) and five pairs of chromosomes in cucumber (RAR 930024) (Figure 2f) exhibited the interstitial type of 45S rDNA organization. This result was supported by previous studies in which P90 and RAR 930024 exhibited two and ten loci of 45S rDNA located in the interstitial region [19,21]. The 45S rDNA loci in melon (P90) and cucumber are located next to the centromeric repeat of melon (Cmcent and Menolird18) and cucumber (Type III) [21,47]. Therefore, this interstitial type of 45S rDNA organization is only found in cucumber and melon P90 but not in *C. hystrix*, *C. debilis*, and another *C. melo* accession (US205). Compared with other *C. hystrix*, *C. debilis*, and *C. melo* (US205), which showed terminal orientations, the location of 45S rDNA in the terminal–interstitial regions in melon (P90) and all interstitial regions in cucumber (RAR 930024) may be supported by transposable element activity during the divergence of cucumber and melon from the same common ancestor 10 Mya [14]. In addition, transposable elements are associated with rDNA and responsible for its chromosomal distribution [57,58]. The insertion of transposable elements (Menolird18) into 45S rDNA and their chromosomal distribution have been demonstrated by Setiawan et al. [21], who reported the distinct discrimination and preferential insertion between melon and cucumber. *C. debilis* has a terminal type of 45S rDNA chromosomal organization. However, whether 45S rDNA loci are located next to centromeric repeats, such as in cucumber, is unclear. Future studies that are similar to those conducted on melon are necessary to find the correlation between 45S rDNA and centromeric repeats in *C. debilis* [59].

Based on our and Sebastian et al.’s [14] phylogenetic results, cucumber is a sister species to *C. hystrix* and closely related to *C. debilis*, and all three were grouped in the phylogenetic tree (Figure 3). This result also is supported by previous studies [4,14,16,17,18]. Interestingly, these three species are not only grouped in one cluster based on their ITS and chloroplast sequences, but are also supported by their 45S rDNA locus number (more than four loci). *C. hystrix* and *C. debilis* have six loci of 45S rDNA compared with the ten loci in cucumber (Figure 2). These results suggest that diploid *Cucumis* species, which are closely related to *C. sativus*, have more than four loci of 45S rDNA. Meanwhile, the species close to *C. melo* have two pairs of 45S rDNA genes. Thus, we hypothesized that the common ancestor of *C. debilis*, *C. sativus*, and *C. hystrix* has 6 45S rDNA loci, and the number of 45S rDNA loci multiplied to 8–10 loci after the divergence. This hypothesis is supported by the finding of non-long-terminal repeat retrotransposons (Menolird18) inserted into the 45S rDNA of cucumber, where this retrotransposon was consistently colocalized with all 45S rDNA loci [21]. In addition, transposable elements contribute to the high variation of rDNA loci in tomato [60]. Similarly, a high variation in cucumber 45S rDNA loci may be caused by transposable elements and/or ectopic recombination events during their domestications [61]. The advantage of high numbers of 45S rDNA loci in cucumber can serve as good markers for precise karyotyping of *Cucumis sativus* together with the use of 5S and/or type I probe [20,47].

Cucumber has a narrow genetic base [62]. Interspecific hybridization can be applied, instead of intraspecific crossing, to increase its genetic base. This approach has been applied in the development of stable allotetraploids of *C. hytivus* (*C. sativus* × *C. hystrix*) [63,64,65,66]. A similar approach can be conducted for the genetic improvement of cucumber through the crossing of cucumber and *C. debilis*. In conclusion, *C. debilis* is an herbaceous plant, that has an oval fruit shape, green fruit peel, and a protrusion-like white spot at the immature stage. This species also has 2n = 2x = 24 chromosomes, similar to the majority of *Cucumis* species. *C. debilis* has three pairs of 45S rDNA loci similar to *C. hystrix*. Based on molecular cytogenetics and phylogenetic tree evidence, *C. debilis* is more closely related to *C. sativus* than *C. melo*. Thus, *C. debilis* can serve as a potential parental line for genetic improvement of cucumber through interspecific hybridization.

## 4. Materials and Methods

### 4.1. Plant Materials

Six *Cucumis* accessions were used in this study (Table 1). *C. debilis* (VN 181-2), *C. hystrix* (LA 09-106), and melon “P90,” derived from Vietnam, Laos, and Japan, respectively, were retrieved from the core collection of Okayama University. Cucumber “RAR 930024” was retrieved from the Institute of Vegetable and Floriculture Science, National Agriculture and Food Research Organization, Japan. *C. metuliferus* (PI 292190) and melon “US205” (PI 182952) were retrieved from USDA-Genebank, USA. The seeds were germinated in a moistened filter paper, transferred to a pot tray, and maintained in a growth chamber.

### 4.2. Chromosome Preparations

The flower buds and root tips were pre-treated with modified Carnoy’s solution II and fixed with 3:1 (*v*/*v*) ethanol:acetic acid in accordance with the work of Setiawan et al. [67]. The root tips and flower buds were washed in an enzyme buffer mixture of 100 mM citric acid and 100 mM sodium citrate at pH 4.8 for 10 min. Then, these samples were macerated in an enzyme mixture containing 1% pectolyase Y-23 (Kyowa Chemical, Osaka, Japan), 2% pectinase (Sigma, St. Louis, MI, USA), and 4% Cellulase Onozuka RS (Yakult, Fountain Valley, CA, USA) at 37 °C for 1 h. The enzymes surrounding the preparate were cleaned, and 60% of acetic acid was dropped and covered with cover slip. The slides were tapped using a probe needle, squashed, and flame-dried for several seconds. The slides were frozen at −81 °C overnight. The cover slips were removed using a razor blade, and the slides were washed with MilliQ water.

### 4.3. Probe Preparation and FISH

The FISH was conducted as described by Setiawan et al. [21]. In brief, the 45S rDNA of the wheat (pTa71) DNA probe was labeled with biotin-nick translation mix (Roche) in accordance with the manufacturer’s protocols. The pretreatments were applied prior to hybridization. The slides were treated with RNase A mixture containing 10 mg/mL RNase A, 20× saline sodium citrate (SSC), and sterile distilled water (SDW) at 37 °C for 1 h. Then, the slides were treated with a pepsin solution containing 500 µg/mL pepsin, 6 N hydrochloride acid, and SDW at 37 °C for 1 h. Each pretreatment was subsequently followed by washing the slides in 2× SSC for 2 min and air dried. Afterward, the slides were re-fixed with 1% paraformaldehyde for 10 min at room temperature, washed in 2× SSC for 2 min, and air dried. The slides were hybridized with a 10 µL 45S rDNA hybridization cocktail consisting of 5 µL formamide, 2 µL 50% dextran sulfate, 1 µL 20× SSC, and 2 µL probe, covered with a cover slip, and sealed with rubber cement at the edges of the cover slip. Thereafter, the slides were incubated in a humidity chamber at 37 °C overnight. Then, the slides were washed in 2× SSC for 2 min, transferred to MilliQ water for 1 min, and air dried. Probe detection was conducted by applying 126 µL detection solution containing 1% bovine serum albumin in 4× SSC and 0.5 µg/mL biotinylated streptavidin–fluorescein isothiocyanate (vector laboratories) onto the slides and incubating them at 37 °C for 30 min. Finally, the slides were washed in 2× SSC for 2 min, transferred into MilliQ water for 1 min, and air dried. Afterward, the slides were counter-stained with 4,6-diamidino-2-phenylindole (DAPI), mounted in a vectashield antifade solution (Vector Laboratories), and observed under a fluorescence microscope (Olympus BX53) equipped with a cooled charge-coupled device camera (Photometrics Cool-SNAP MYO). The FISH images were processed using Metamorph, Metavue imaging series version 7.8 and edited with Adobe Photoshop CS6.

### 4.4. Phylogenetic Analyses

Phylogenetic analysis was performed as described in the work of Sebastian et al. [14]. The ITS of 45S rDNA and chloroplast sequences (trnL intron, trnL-F, rpl20-rps12, trnS-trnG intergenic spacers, rbcL, and matK genes) of six *Cucumis* species were retrieved from the National Center for Biotechnology Information database. Table 2 lists the details of the sequences. The sequences were aligned using MUSCLE [68], and phylogenetic analysis was performed using the maximum likelihood method based on the work of Tamura et al. [69] with 1000 bootstrap replicates in MEGA7 [70].

## Figures and Tables

**Figure 1 plants-11-01960-f001:**
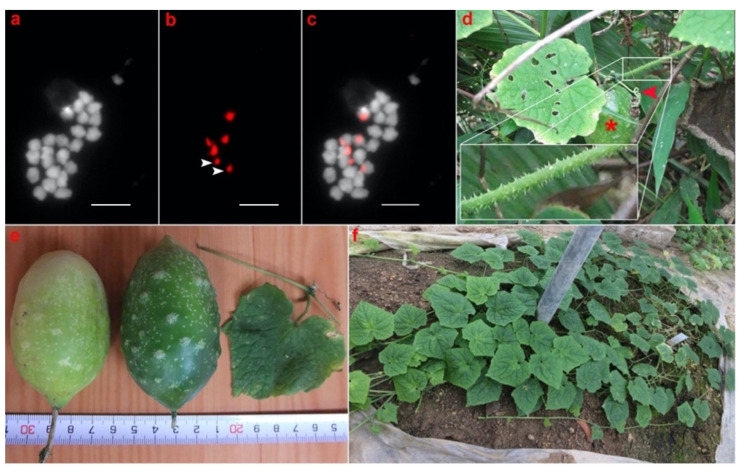
Chromosomal distribution of 45S ribosomal DNA (rDNA) in *Cucumis debilis* and its morphological characters. *C. debilis* chromosomes stained with 4,6-diamidino-2-phenylindole (DAPI) (**a**); 45S rDNA signals (red) detected with anti-dig rhodamine (**b**); physical mapping of 45S rDNA in *C. debilis* chromosomes (**c**); *C. debilis* plant found in Vietnam successfully produced fruits (asterisk) in its habitat (**d**); and had sparsely aculeate tendrils (red arrowhead) for climbing and hairy stem (inset), deeply cordate leaf base, broad triangular leaf lobes with spiny hairs, and acuminate leaf apex, the shape of the fruit is oval with the green-colored immature fruit peel (center) and early mature fruit peel color with a definite break in color from green to slightly yellow (left) and leaf morphology with shallow leaf lobe (**e**); *C. debilis* plant grown and maintained in Okayama University greenhouse and failed to develop flowers and fruits (**f**). Arrow heads depict the minor signals of 45S rDNA. Scale bar in (**c**) = 10 μm.

**Figure 2 plants-11-01960-f002:**
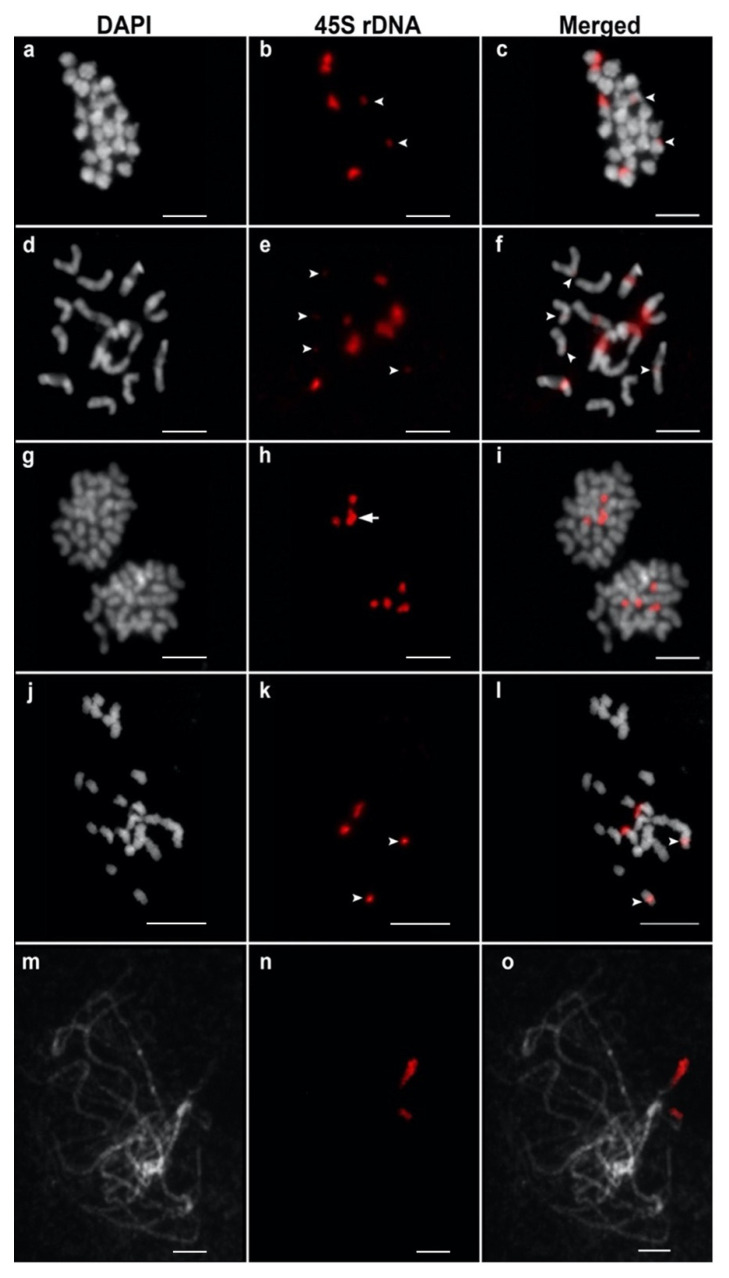
Physical mapping of 45S rDNA on mitotic and meiotic chromosomes of *Cucumis* species. *C. hystrix* (**a**–**c**), *C. sativus* “RAR 930024” (**d**–**f**), *C. metuliferus* “US 143” (**g**–**i**), *C. melo* “P90” (**j**–**l**), and *C. melo* “US205” (**m**–**o**) were stained with DAPI and subjected to fluorescent in situ hybridization (FISH) with the 45S rDNA probe (red). Arrow heads depict the minor signals of 45S rDNA. Arrow depicts the overlapped 45S rDNA loci. Scale bars = 10 μm.

**Figure 3 plants-11-01960-f003:**
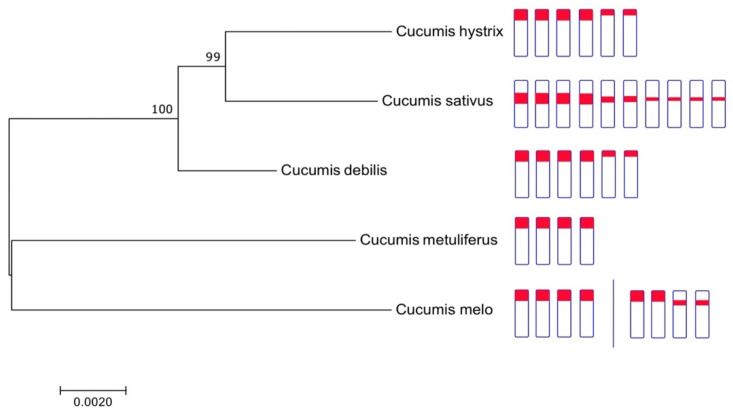
Phylogenetic tree of *Cucumis* species based on the combined sequences of nuclear internal transcribed spacer (ITS) and chloroplast followed by chromosome ideograms constructed from FISH images. The evolutionary history was inferred by using the maximum likelihood method. The tree with the highest log likelihood (−7753.85) is shown. Likelihood bootstrap values are given at the nodes. Evolutionary analyses were conducted in MEGA7.

**Table 1 plants-11-01960-t001:** Species origin and chromosomal location of 45S rDNA in several *Cucumis* accessions.

Accession	Species	Origin	Chromosome Number (2n)	Number of 45S rDNA Pairs		45S rDNA Location
				Major	Minor	
VN 181-2	*C. debilis* L.	Vietnam	24	2	1	Terminal
LA 09-106	*C. hystrix* L.	Laos	24	2	1	Terminal
RAR 930024	*C. sativus* L.	USSR (Uzbekistan, Kazakhstan, and Kyrgyzstan area)	14	3	2	Interstitial
P90	*C. melo* L.	Japan	24	1	1	Terminal & Interstitial
US 205	*C. melo* L.	India	24	2	0	Terminal
US 143	*C. metuliferus* L.	South Africa	24	2	0	Terminal

**Table 2 plants-11-01960-t002:** Accession numbers of ITS nuclear and chloroplast sequences used in this study.

Species	ITS Spacer	*trnL* Intron	*trnL-F* Spacer	*rpl20-rps12* Spacer	*trnS-G* Spacer	*rbcL* Gene	*matK* Gene
*C. debilis*	HM596905	HM597013	HM597012	HM596962	HM597068	KY434400.1	KY458068.1
*C. hystrix*	HM596908	HM597017	HM597016	HM596965	HM597076	DQ785832	DQ785846
*C. sativus*	HM596928	HM597036	HM597036	HM596984	HM597104	DQ5357471.1	DQ536662.1
*C. melo*	HM596916	HM597025	HM597025	HM596973	HM597086	DQ535800	DQ536659
*C. metuliferus*	EF093517	DQ785876	DQ785876	DQ785862	HM597088	DQ785834	DQ785848

## Data Availability

The data presented in this study are available within the article.

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
