# Peer review of "The Divergence of Chromosome Structures and 45S Ribosomal DNA Organization in Cucumis debilis Inferred by Comparative Molecular Cytogenetic Mapping"

_plants, 2022, doi:10.3390/plants11151960_

Round 1

Reviewer 1 Report

  1. In the introduction section, avoid writing genus names everywhere. You can write C.melon
  2. Result section 2.4: explain the chromosome ideograms mentioned in figure 3. 
  3. Line 180 and 185: explain the results with previous studies.
  4. Line 187-189: Rewrite the sentence “Therefore, this interstitial…….”
  5. Line 55, 155, 158, 196: please mention the author’s name.
  6. Format figure legends according to the journal format
  7.  Be consistent in writing figure no. throughout the manuscript.
  8. Spelling and grammar;

Line 24: and protrusion-like white spots during the immature fruit stage

Line 63: confer

line72: compared to other

line 95: spots 

line 185: This result was supported by previous studies

line 203: all three were grouped in

Reviewer 2 Report

Revision of the --Manuscript Draft-- Manuscript Number: plants-1789467

Title: The divergence of chromosome structures and 45S ribosomal DNA organization in Cucumis debilis inferred by comparative molecular cytogenetic mapping

Type: Article

This is a manuscript within the scope of the journal "Plants (ISSN 2223-7747)". It is relevant for providing novel information of not well-studied Cucumis debilis species as an herbaceous plant, with an oval fruit shape, green fruit peel, and a protrusion-like white spot at the immature stage. Based on molecular cytogenetics and phylogenetic tree evidence, C. debilis is more closely related to C. sativus than C. melo. Thus, C. debilis can serve as a potential parental line for genetic improvement of cucumber through interspecific hybridization.

The manuscript is scientifically sound and the experimental design is appropriate to test the hypothesis. However, I suggest that the actual document must be improved for better understanding. For example, authors need to change the order of the Materials and methods section as the second one after the Introduction section. Moreover, they must mention details about morphological characters determined by C. debelis materials. Consequently, for the importance and novelty they need to write data on Tables and Figures regarding the morphological traits recorded in the study and not only the results that appear in Figure 1.

For the last aspect, I suggest reading the papers:

Chikh-Rouhou, H., Mezghani, N., Mnasri, S., Mezghani, N., & Garcés-Claver, A. (2021). Assessing the genetic diversity and population structure of a tunisian melon (Cucumis melo L.) collection using phenotypic traits and SSR molecular markers. Agronomy, 11(6), 1121. https://doi.org/10.3390/agronomy11061121

Merheb, J., Pawełkowicz, M., Branca, F., Bolibok-Brągoszewska, H., Skarzyńska, A., Pląder, W., & Chalak, L. (2020). Characterization of lebanese germplasm of snake melon (Cucumis melo subsp. melo var. flexuosus) using morphological traits and SSR markers. Agronomy, 10(9), 1293. https://doi.org/10.3390/agronomy10091293

Please, I suggest the authors to follow that in the text, reference numbers should be placed in square brackets with same font [ ], and placed before the punctuation.

The actual manuscript needs to be corrected to minor text editing. I suggest the authors for improving the ms to take care for the whole document e.g. LNs 38-41, 44, 45, 52-54, 57, 93, 97, 98, 108-110, 113, 115-117, 126, 128. 130-132, 138, 155, 158, 174, 179, 183, 184, 196, 204, 208, 210, 282.

In Figure 1, please, see the text in LNs 80, 81. In general, I suggest the authors to elaborate again d) - f) and draw the scale bar for all pictures. Please, in the photo e) I suggest (if possible) to change it without a ruler.

In Figure 2, see the LNs 101-103.

I suggest the authors draw the scale bar for all pictures. Please, in the photo e) I suggest (if possible) to change it without a ruler.

In the Reference list, the DOI numbers (Digital Object Identifier) are not mandatory but highly encouraged.

Please, more aspects are mentioned in the revised ms and attached document.

With regards

Round 2

Reviewer 1 Report

The revised version of the MS looks appropriate. The authors have addressed all the comments from the reviewers. I recommend acceptance of the MS.

Reviewer 2 Report

Dear colleagues,

I accurately revised the second version of the manuscript entitled: The divergence of chromosome structures and 45S ribosomal DNA organization in Cucumis debilis inferred by comparative molecular cytogenetic mapping.

I believe it has been sufficiently improved to warrant publication in Plants (ISSN 2223-7747).

However, I suggest the authors consider again in Figure 1 the scale bar for photos d, e, and f. In the actual view, it will be confusing for the readers.

Moreover, in LN 121 it is necessary to correct as ¨Scale bar in (a, b, c)¨.

With regards